

# Accounting for the effects of non-ideal minor structures on the optical properties of black carbon aerosols

Shiwen Teng[1,2], Chao Liu[1,2,*], Martin Schnaiter[3], Rajan K. Chakrabarty[4], Fengshan Liu[5]

[1]Collaborative Innovation Center on Forecast and Evaluation of Meteorological Disasters, Nanjing University of Information
Science & Technology, Nanjing 210044, China
[2]Key Laboratory for Aerosol-Cloud-Precipitation of China Meteorological Administration, School of Atmospheric Physics,
Nanjing University of Information Science & Technology, Nanjing 210044, China
[3]Karlsruhe Institute of Technology, Institute of Meteorology and Climate Research, 76021 Karlsruhe, Germany
[4]Center for Aerosol Science and Engineering, Department of Energy, Environmental and Chemical Engineering, Washington
University in St. Louis, Missouri - 63130, USA
[5]Black Carbon Metrology, Measurement Science and Standards, National Research Council, Ottawa, Ontario K1A 0R6,
Canada

*Correspondence to*: Chao Liu (chao_liu@nuist.edu.cn)

**Abstract.** Black carbon (BC) aerosol is the strongest sunlight-absorbing aerosol, and its optical properties are fundamental to
radiative forcing estimations and retrievals of its size and concentration. During incomplete combustion, BC particles exist
as aggregate structures with small monomers, and are widely represented by the idealized fractal aggregate model formed by
monodisperse spherical monomers in point-contact. In reality, BC particles possess complex and non-ideal minor structures
besides the overall aggregate structure, altering their optical properties in unforeseen ways. This study introduces a
parameter 'volume variation' to quantify and unify different minor structures, and develops an empirical relation to account
for their effects on BC optical properties from those of ideal aggregates. Minor structures considered are the polydispersity
of monomer size, irregularity and coating of individual monomer, and necking and overlapping among monomers. The
discrete dipole approximation is used to calculate the optical properties of aggregates with these minor structures. Minor
structures result in scattering cross section enhancement slightly more than that of absorption cross section, and their effects
become weaker with the increase of wavelength. Their effects on the angular-dependent phase matrix as well as asymmetry
factor are negligible. Our results suggest that a correction ratio of 1.05 is necessary to account for the mass/volume
normalized absorption and scattering of non-ideal aggregates in comparison to ideal ones. In other words, minor structures
tend to enhance the BC mass absorption and scattering by 5 %, which also applies to aggregates with multiple minor
structures. With the conclusions, the simulations of optical properties of non-ideal aggregates are greatly simplified, because
they can be directly obtained from those of the corresponding ideal aggregates. We expect this generalized correction to find
wide use for modeling realistic BC aggregates due to the simplicity involved in generating ideal fractal aggregates, and to be
valuable for not only interpretation of measurements but also practical modeling that requires large amount of simulations.



## 1 Introduction

Black carbon (BC), produced by incomplete combustion of fossil- and biofuels, and biomass, is one of the strongest sunlight-absorbing atmospheric constituents (Jacobson, 2001; Andreae and Gelencsér, 2006). BC particles affect the radiative heat balance at global and regional scales by absorbing solar radiation and reducing the radiation reaching the

surface (Crutzen and Andreae, 1990; Menon et al., 2002; Kahnert and Devasthale, 2011). Thus, the optical properties of BC particles are fundamental not only to radiative forcing estimations but also to retrievals of their size and concentration, whereas they are in turn highly dependent on the complex and heterogeneous morphology of BC particles.

The fractal aggregates have been widely used to represent BC geometries and to obtain their optical properties (Farias et al.,

1996; Liu et al., 2012; Bescond et al., 2013). In the fractal aggregate model, aggregates are formed by numerous perfect same-sized spheres, also called monomers, that are in point-contact. Mathematically, fractal aggregates are described by the following statistic scaling rule (Sorensen, 2001):

$$N = k_f \left(\frac{R_g}{a_o}\right)^{D_f},\tag{1}$$

where $N$ is the number of monomers in an aggregate, $a_o$ is the radius of primary monomers, and $R_g$ is the gyration radius. $k_f$

(fractal prefactor) and $D_f$ (fractal dimension) are parameters describing the aggregate overall structure, i.e., lacy or compact. Numerous experimental evidence confirms that freshly emitted BC particles are normally chain-like and lacy aggregates with $D_f$ less than 2, whereas aged BC aggregates tend to be more compact with $D_f$ values even close to 3 (Sorensen, 2001; Chakrabarty et al., 2009; Wang et al., 2017).

BC particles in the ambient atmosphere show significant diversities in their morphologies, and are much more complex than the idealized fractal-like aggregates specified by Eq. (1), which only describe their general morphology (Wu et al., 2015a; Pirjola et al., 2017). Figure 1 shows some examples of transmission/scanning electron microscope (TEM/SEM) images of BC particle sampled under different real-world scenarios (Gwaze et al., 2006; Kamimoto et al., 2007; Chakrabarty et al., 2009; Yon et al., 2015; Wang et al., 2017). The particles are obviously highly complex, not only in their overall morphology,

but also concerning their detailed structures. As is apparent, actual BC particles display features that are quite different from the assumptions used for fractal aggregates (e.g., same-sized, perfect spherical, point-contacting), such as polydispersity of monomer size, and necking and overlapping among monomers. Furthermore, BC aggregates also get mixed with organic or inorganic aerosols in varying proportions during transport and aging (e.g., the left panels of Fig. 1), and the mixing can alter their morphology as well. Several studies have been carried out to quantify these detailed structures to better represent them

in numerical models (Cheng et al., 2014; Moteki, 2016). Among them, the polydispersity of monomer size is the most well studied (Chakrabarty et al., 2006, 2007; Bescond et al., 2014; Wu et al., 2015b), and a complete set of methods have been established to characterize monomer size distribution. Bourrous et al. (2018) developed a semi-automatic analysis to obtain





the overlap coefficient and specific surface areas of aggregates. All these studies show clearly and quantitatively the existence of detailed minor structures of actual BC aggregates.

The overall fractal aggregate morphologies and their influence on the optical properties of BC particles have been well
studied (Sorensen, 2001; Liu and Mishchenko, 2005; Kahnert and Devasthale, 2011). The minor structures mentioned above, which have even smaller length scales than the aggregate overall size, also gain significant attentions for their influences on aggregate optical properties (Farias et al., 1996; Bond et al., 2006; Scarnato et al., 2013). As a consequence, more realistic and detailed models have been developed to improve our knowledge on the optical properties of BC aggregates. Farias et al. (1996) confirmed the effects of particle polydispersity on mean optical cross sections, with an amplification ratio of 1.2 and
1.8 for absorption and scattering, respectively; however, a recent study by Liu et al. (2015) found the ratios to be 1 and 2.5, respectively, with the standard deviation of 1.5 for particle size distribution. Bescond et al. (2013) evaluated the impact of overlapping and necking on the radiative properties of soot aggregates, with a main focus on their effects on the depolarization. Skorupski and Mroczka (2014) extended the investigation to the effects of overlapping and necking on the absorption and scattering properties, and found that the effects of 'small' necking can be ignored, but when overlap occurs,
the effects are pronounced. However, Yon et al. (2015) showed that overlapping and necking may significantly affect the absorption and scattering properties with the amplification factor being up to 2 at 266 nm. Meanwhile, small to moderate scale coating in different forms has been considered in numerical models to study their effects on the optical properties, and the impact of coating on absorption was found to be 1.1 averagely in the visible spectral range (Scarnato et al., 2013; Dong et al., 2015; Doner et al., 2017). For relatively small amount of coating, the extinction and absorption cross sections are
approximately enhanced respectively by 5 % and 3 % as the volume fraction of coating increases from 0.01 to 0.2, as shown by Liu et al. (2012). However, Dong et al. (2015) indicated that the amplification factors of absorption and scattering increase to 1.15 and 2 for partially coating with volume fraction of 0.5.

Table 1 summarizes some of the previous studies investigating the effects of non-ideal minor structures on BC optical
properties (Farias et al., 1996; Liu et al., 2012; Scarnato et al., 2013; Skorupski and Mrocz, 2014; Yon et al., 2015; Dong et al., 2015; Doner and Liu, 2017; Doner et al., 2017). Key parameters for simulations, including the monomer number, wavelength, and characteristic scale of structures, and the key conclusions are given in the table. The average amplification factor of optical properties caused by these minor structures is used to specify the effects. Obviously, quite different conclusions are found, and this can be attributed to the different assumptions and parameters used in the numerical models.
Because of the lack of a unified approach and criterion to quantify the effects of various minor structures, it is difficult to gain an overall quantitative understanding of such effects and to incorporate these effects in numerical models for climate modeling and retrieval of BC concentrations in optical diagnostics. The relevant investigations will also likely continue to focus on the effects of certain specific structures. Furthermore, with a clear knowledge that the minor structures of aggregates do influence their optical properties, the numerical simulations and applications for radiative properties of BC





particles become more difficult, because more sophisticated and less computationally efficient numerical models are required to accurately capture the effects of such non-ideal structures. Thus, it becomes an open question on how we should deal with these different structures and apply them to practical applications in the future.

This study systematically investigates the effects of several minor structures on the optical properties of BC aggregates, including the polydispersity of monomer size, monomer surface irregularity, coating, necking, and overlapping, and develops a simple method to account for their effects from the optical properties of the corresponding idealized fractal aggregates. This paper is organized as follows. Section 2 introduces the numerical models of BC aggregates with different minor structures, and unifies them by introducing a parameter of 'volume variation'. The effects of the minor structures and

the empirical relationship to consider them in practical applications are discussed in Sect. 3. Section 4 concludes this study.

## 2 Minor structures

Actual BC aggregates always possess imperfect detailed structures, and, to account for their effects on the optical properties of BC aggregates in numerical simulations, accurate numerical models to adequately represent such structures are needed. We define the imperfect geometries, such as monomer size polydispersity, monomer non-sphericity (or irregular monomer

surface), thin coating, necking, and overlapping, as "minor structure" in this paper. First, the "minor" indicates that the structures are in the monomer scale, and don't strongly alter the overall fractal aggregate structure. Secondly, the structures are assumed to change the aggregate total volume/mass slightly, e.g. a difference less than 20 % from the idealized aggregate. The following introduces the five minor structures that will be considered in this study.

The monomers are often assumed as same-sized spheres for simplification in numerical studies on BC optical properties. However, measurements of sampled BC particles reveal that BC monomers have clear variations on their sizes, ranging from approximately 10 nm up to even 100 nm (Dankers and Leipertz, 2004; Chakrabarty et al., 2006; Bescond et al., 2014). Thus, the polydispersity of monomer size is defined as the first minor structure, i.e., $M_1$. Based on various observations (Köylü and Faeth, 1994; Lehre et al., 2003), a lognormal size distribution provides a close representation on realistic monomer size

distribution:

$$P(a_1) = \frac{1}{2a_1\sqrt{2\pi}lnc_1}exp\left[-\left(\frac{ln(a_1/a_o)}{\sqrt{2}lnc_1}\right)^2\right],\tag{2}$$

where $a_o$ and $c_1$ are the geometric mean radius and geometric standard deviation, respectively, and $a_1$ indicates the radii defined for $M_1$. The $M_1$ shown in Fig. 2 schematically gives the two-dimensional illustration of aggregates with polydisperse monomers. As aggregates size $N$ becomes large, the monomer sizes should follow the aforementioned size distribution. We

fix $a_o$ as that of the same-sized monomers, and the geometric standard deviation $c_1$ is the only variable to specify the polydispersity. For $c_1$ larger than 1, the monomer radii distribute over a wider range. More details on the definition of the monomer size distribution can be found in Liu et al. (2015).



Secondly, the actual BC monomers in the atmosphere would never be perfectly spherical, and there must be irregularity or surface roughness to some degree on BC monomers. Therefore, we consider the irregularity of BC monomers as the second structure feature M$_2$ in this study. Various numerical models, either stochastic or regular, have been developed to represent

the rough surface of aerosols or ice crystals (Li et al., 2004; Kahnert et al., 2012a), and some of them can also be adopted for BC monomers. The surface roughness is a morphological feature with random nature, and the model introduced by Muinonen et al. (1996) is used for each BC monomer. The model uses the multivariate lognormal statistics (Gaussian random shape), and the monomer surface can be expressed by:

$$a_2(\vartheta, \varphi) = \frac{a_o}{\sqrt{c_2}} exp\big(s(\vartheta, \varphi)\big).$$ (3)

Similarly, $a_2(\vartheta, \varphi)$ is the function to define monomer surface in model M$_2$. $\vartheta$ and $\varphi$ are the azimuth and elevation angles for given spherical coordinates. $s(\vartheta, \varphi)$ is the random variable obeying the Gaussian distribution with the mean value of 0 and a fixed variance of 0.05, and the same definitions are given in Muinonen et al. (1996). The degree of roughness or irregularity of each monomer is directly controlled by the standard variance of Gaussian distribution, i.e., parameter $c_2$. With the decrease of $c_2$, the surface of monomers becomes more rough, and spikes on the surface will be sharper. Following Eq. (3),

an example of M$_2$ for two monomers is illustrated in Fig. 2, and the spikes on monomer spheres can be either outward or inward due to its randomness. The changes in the aggregate volume due to irregularity from that of the perfect aggregate are clearly represented by the red region in Fig. 2.

Once emitted into the atmosphere, BC particles are unavoidably mixed with non- or weakly-absorbing aerosols, such as

sulfates, nitrates, and organic carbon, which significantly affects the absorption and scattering properties of BC particles (Jacobson, 2000; Schwarz et al., 2008; Liu et al., 2013). The third minor structure (M$_3$) considered is coating. The mixing states are diverse under different situations, so various complex coating models have been developed (Scarnato et al., 2013; Dong et al., 2015). Since this study focuses on monomer minor structures, a thin coating model, i.e., M$_3$ shown in Fig. 2, is considered by wrapping each monomer with a spherical coating shell. Then, the radius of the coated sphere can be described

by a coating parameter $c_3$:

$$a_3 = c_3 \cdot a_o,$$ (4)

where $c_3$ is a dimensionless parameter greater than 1. The red region of M$_3$ in Fig. 2 indicates the coating material. As $c_3$ increases, the coating becomes thicker and its fraction also increases. Note that we consider only non-absorbing materials as coating in this study, because absorbing aerosols will introduce addition complication.

Besides the minor structures for individual monomer discussed above, structures between neighboring monomers also exist and affect the radiative properties of BC aggregates. The fourth minor structure (M$_4$) is defined as "necking" between neighboring monomers, which is resulted from sintering processes or surface growth around the contact point (Zaitone et al.,



2009; Bescond et al., 2013). We introduce one of the simplest necking forms, i.e. cylindrical connector, and, in other words, cylinders are added around the contact point of two neighboring monomers along their centers. Again, a dimensionless parameter $c_4$ is used to quantify the radius of the circular cylinders $a_4$:

$$a_4 = c_4 \cdot a_o \,. \tag{5}$$

The red region of $M_4$ in Fig. 2 shows the necking between two neighboring monomers. When $c_4=0$, the connection does not exist, and, for $c_4=1$, the radius of the cylinder is equal to the monomer radius.

The last minor structure ($M_5$) considered is overlapping between two connected monomers. Two general ways are used to generate overlapping: one is to reduce the distance between the centers of two connected monomers (Brasil et al., 1999; Yon
et al., 2015) that will influence the aggregate overall compactness significantly; another is to enlarge the radius of primary monomers but keep their centers unchanged (Doner and Liu, 2017), and the aggregate overall compactness is less influenced. To keep the morphology of aggregates unchanged and consistent with that of other minor structures mentioned above, we generate the overlapping using the latter way in this paper. The enlarged monomer radius is expressed as:

$$a_5 = c_5 \cdot a_o \,. \tag{6}$$

When $c_5=1$, the monomers are in point-contact as in the ideal fractal aggregates. With the increase of $c_5$, the monomers become larger and there is severe overlapping between connected monomers. Although overlapping leads to a decrease in the aggregate volume as shown in the red region of $M_5$ in Fig. 2, the total volume still increases as the volume of each monomer is larger.

To summarize, the top panels of Fig. 2 schematically illustrate the definitions of the five numerical models in two dimension. With these basic minor structures defined quantitatively, it is straightforward to build fractal aggregates with these structures. First, idealized fractal aggregates with same-sized spheres in point-contact are generated using a tunable cluster-cluster aggregation algorithm (Filippov et al., 2000). From the generated ideal aggregate, we modify each monomer with different minor structures as discussed above to develop the non-ideal aggregates. The bottom panels of Fig. 2 show some examples
of aggregates with minor structures, which are modified from ideal aggregates generated using fractal dimension and fractal prefactor of 1.8 and 1.2, respectively (Sorensen and Reborts, 1997; Sorensen, 2001; Kahnert and Devasthale, 2011). The aggregates with only 15 monomers are displayed to better highlight the details of minor structures in each case.

Geometrically, different numerical models have to be developed for different minor structures, which make direct
comparisons of these different minor structures challenging since these structures can lead to different effects on the optical properties and their effects are seemingly not directly comparable. Only if these minor structures were defined in a unified manner, their influences on the BC optical properties can be quantitatively compared and the nature of their effects can be better understood. For the five minor structures considered, five different dimensionless parameters are used to quantify them independently. To this end, we have to find a unified parameter that includes the effects of all the five structures to



quantify them. Considering that all minor structures influence not only the geometry but also particle overall volume/mass, we convert geometrical differences into volume/mass differences in this study. Thus, a unified parameter $\beta$, called the volume variation, is defined, and it simply represents the relative volume difference between an aggregate with minor structures and the corresponding perfect one:

$$\beta = \frac{V_\beta - V_o}{V_o}, \tag{7}$$

where $V_o$ is the volume of idealized fractal aggregates, and $V_\beta$ is the volume of aggregates with minor structures. Because the particle volume plays one of the most primary effects on particle optical properties, especially for particles much smaller than the incident wavelength (e.g. single monomer), the parameter of volume variation will not only unify the definition of the minor structures for comparison but also provide a simple quantitative method to account for their effects on BC optical properties.

Figure 3 illustrates the relationships between the volume variation $\beta$ and different structure parameters. The volume of aggregates with minor structures can be calculated numerically in a discretized space domain with a sufficiently high spatial resolution, which will also be used for optical property simulations in the following section. Obviously, the total volume variation $\beta$ varies rapidly with the change of structure parameters, and these structure parameters under different volume variations can be uniquely identified with the clear relationships. When $\beta$ is 0, the BC aggregates are the original ideal ones. The volume variations range from 0 to 0.2 are considered in this paper. The relationship between the volume variation and the parameters at a given level of volume variation is also illustrated in Table 2, in which the five dimensionless parameters leading to the given $\beta$ values are listed. It is noticed that these values correspond to conditions that only one minor structure is allowed in each case. Meanwhile, the relationship may vary slightly for different aggregates due to their irregular natures, so the $\beta$ value of each aggregate will be calculated in the following section.

## 3 Effects of minor structures

We use the discrete dipole approximation (DDA) method to simulate the optical properties of BC aggregates with those minor structures, because it is highly flexible in term of defining particle shape and composition (Yurkin and Hoekstra, 2007; Kahnert et al., 2012b). In the DDA simulations, the scatterers are discretized into numerous small sub-volumes, namely dipoles, and particles with complex shapes can be accurately described via such dipoles as long as the diploes are sufficiently small in comparison to the wavelength considered. The ADDA code developed by Yurkin and Hoekstra (2007; 2011) is used. This paper mainly discusses the optical properties at an incident wavelength of 500 nm, and the BC refractive index is assumed to be 1.8+0.6$i$. The original monomer radius in ideal aggregate is assumed to be $a_o$ = 15 nm. To ensure that the detailed monomer structures are adequately represented in the DDA simulations, we discretize the particles using 200 dipoles per wavelength, and this leads to over 900 dipoles per monomer. For the overall geometry, we consider the fractal





dimension values obtained from Wang et al. (2017), and build both lacy aggregates ($D_f$=1.8) and aged ones with compact structures ($D_f$=2.3) for optical studies.

Figure 4 compares the relative differences (RDs) between optical properties of the idealized aggregates and those with

different minor structures as a function of volume variation $\beta$. Here, aggregates with 200 monomers are considered, and the top and bottom panels are for results of aggregates with $D_f$=1.8 and $D_f$=2.3, respectively. Positive RDs mean the minor structure enhance the corresponding optical properties. With the increase of volume variation, the RDs on the extinction and absorption cross sections increase almost linearly, and different minor structures tend to have obvious enhancement in extinction and absorption cross sections to varying degrees. The effects of necking are largest with ~ 35 % RDs as $\beta$

increases to 0.2, while those of coating are smallest with the RDs remain less than 10 %. The RDs of single scattering albedo (SSA) also become larger with the increase of $\beta$ (up to ~15 % at $\beta$=0.2), whereas these differences for different minor structures are less significant than those in extinction and absorption cross sections. The results of asymmetry factors for different minor structures display very small differences, and the RDs, less than 2 % even for the largest $\beta$ value considered, are close to zero over the entire range of $\beta$ considered, suggesting that it is nearly unaffected by the type and magnitude of

minor structure. Furthermore, results for lacy and compact aggregates are similar, and only the polydispersity of monomer size shows relatively larger impact on asymmetry factor due to the change of fractal dimension.

With almost no influence on the asymmetry factors, the effects of these minor structures on the angular-dependent scattering matrix elements are also expected to be minor. Figure 5 shows the normalized phase function and two non-zero scattering

matrix elements ($P_{22}$ and $P_{34}$) for aggregates with minor structures at a volume variation of $\beta$=0.1, and those of perfect aggregates are given as a direct comparison. Because minor structures show even less impacts on other non-zero scattering matrix elements ($P_{12}$, $P_{33}$, and $P_{44}$), they will not be shown here. The normalized phase functions and $P_{34}$ of aggregates with different minor structures agree closely with those of idealized aggregates. The polydispersity (the blue lines) has essentially no influence on $P_{11}$ but shows impacts for backward scattering with scattering angles larger than 90°, and the effects are

slightly larger for compact aggregates. For $P_{22}$, the necking causes the largest decreases among the five structures, whereas the overall differences remain small (note that the range of $P_{22}$ values plotted are between 0.96 and 1). To summarize, minor structures have negligible effects on the angular-dependent scattering even at a volume variation of 0.1. Thus, the scattering matrix elements of aggregates will not be further considered and the following discussion will focus only on the integral scattering properties (i.e., those shown in Fig. 4).

Different from perfect aggregates, whose optical properties can be calculated conveniently by accurate models such as the multi-sphere T-matrix method (Mackowski, 2014) and the General Multiple-Mie method (Xu, 1995), modeling the optical properties of aggregates with minor structures is much more tedious. This also makes its applications challenging due to the time-consuming simulations and uncertainties in the definition of minor structure parameters. Thus, it is important and





necessary to find an empirical relationship between optical properties of aggregates with and without minor structures. The relationship, which can estimate the optical properties of aggregates with minor structures from those of idealized ones, should be general and simple for the purpose of practical applications. Fortunately, Figures 4 and 5 indicate that although the minor structures have different influences on aggregates optical properties, they do have some similar features. For example,

their effects on the extinction and absorption cross sections are almost proportional to the volume variation, and the effects on the scattering matrix are ignorable. Those features can be applied for the empirical relationship.

To develop the empirical relationship based on the current DDA results, the significant influence of the volume variation on the cross sections should be considered first. This is reasonable, because minor structures only change monomer shapes. In

the visible and shortwave infrared spectra ranges, BC monomers are normally in the Rayleigh regime, and the corresponding absorption and scattering cross sections are proportional to the volume and volume square, respectively. Considering the linear relationship shown in the absorption cross section in Fig. 4, it is reasonable to approximate the absorption cross section of aggregates with minor structures by:

$$ABS_\beta = B_{ABS} \cdot \frac{V_\beta}{V_o} \cdot ABS_o = B_{ABS} \cdot (1 + \beta) \cdot ABS_o \,, \qquad (8)$$

where $ABS_\beta$ is the absorption cross section of the imperfect aggregate with a volume of $V_\beta$ and volume variation of $\beta$, and $ABS_o$ and $V_o$ are the values of the corresponding perfect aggregate. The $1 + \beta$ term indicates the change in overall volume due to minor structures, and parameter $B_{ABS}$ can be understood as a "correction ratio" indicating the direct effects of minor structures on the absorption cross section. Similarly, the relationship for the scattering cross section can be given as:

$$SCA_\beta = B_{SCA} \cdot \left(\frac{V_\beta}{V_o}\right)^2 \cdot SCA_o = B_{SCA} \cdot (1 + \beta)^2 \cdot SCA_o \,. \qquad (9)$$

Meanwhile, the extinction cross section can be obtained by the sum of the scattering and absorption cross section. It is noticed similar relationships to Eqs. (8) and (9) have been proposed by Farias et al. (1996) to account for the effect of monomer polydispersity, but without the terms $B_{ABS}$ and $B_{SCA}$. Based on the results shown in Figs. 4 and 5, the scattering matrix and asymmetry factor are unchanged. Because the volume of BC, absorbing material, is not changed in the coating case, the volume variation term $(1 + \beta)$ in Eq. (8) is removed in the empirical approximation.

The next step is to determine the values of $B_{ABS}$ and $B_{SCA}$ for different minor structures. We approximate the two correction ratios based on the results shown in Fig. 4, and the results are listed in Table 3. All values in the table are larger than 1, meaning that the minor structures also enhance the equivalent mass absorption and scattering of aggregates. Larger values of $B_{ABS}$ or $B_{SCA}$ mean that the minor structure has a stronger influence. The $B_{ABS}$ of necking is the largest with a value up to

1.08, and the $B_{ABS}$ of polydispersity is the smallest, only 1.02. The correction ratios are all in the range of 1.01 and 1.08 for absorption and scattering, so the empirical relationships in Eqs. (8) and (9) can be further simplified by assigning a constant correction ratio of $B_{ABS} = B_{SCA} = 1.05$ for all cases. In other words, we specify an average enhancement rate on the



scattering and absorption of aggregates with minor structures after considering volume variation. The resulting cross sections can be used to obtain the new single-scattering albedo of aggregates with minor structures as well. Another advantage of the proposed relationships is that only a volume or mass variation is needed for the approximation, whereas evaluation of the geometric parameters of the minor structure is much more challenging.

To evaluate the performance of the empirical relationship, Figure 6 replotted the results of Fig. 4, and directly compares the results using aforementioned correction relationships (with the help of those of perfect aggregates) and those of the rigorous DDA simulations for non-ideal aggregates. To remove the effects of differences on volume in the following evaluations, mass extinction and absorption cross sections (MEC and MAC), i.e., values per unit BC mass, are considered instead of

absolute cross sections, and we use a BC density of 1.8 g cm$^{-3}$ (Bond and Bergstrom, 2006). Thus, the REs in Fig. 6 show the relative errors of empirical relationships on approximating the optical properties of BC aggregates with minor structures. The gray shaded regions indicate REs between ± 5 %. Again, the top panels are for lacy aggregates, and bottom ones are for compact aggregates. Obviously, the relative errors between direct DDA simulations of imperfect aggregates and results from empirical relationships are well under the ± 5 %. This indicates that the empirical relationships can provide a good

approximation on the optical properties of BC aggregates with minor structures, at least for cases considered in this study.

The performance of this simple approximation utilizing the optical properties of perfect aggregates is further evaluated for different-sized aggregates at different wavelengths. Figure 7 compares the optical properties from direct simulations of aggregates with minor structures (markers) and empirical approximations based on properties of perfect aggregates (solid

black lines). Here, we consider only lacy aggregates with a fractal dimension of 1.8, and aggregates containing 50 to 600 monomers are considered. The gray shaded regions indicate ± 5 % relative errors within the results from the empirical relationships. The MEC and MAC for aggregates with monomer polydispersity and necking are slightly smaller and larger than the empirical results, respectively. Nevertheless, all the makers, i.e., results from direct simulations of aggregates with minor structures, lie within the ± 5 % gray shaded areas. The empirical results of SSA agree more closely with direct

simulation results. As expected, the asymmetry factor is least affected by minor structures with average REs less than 2 %. The monomer polydispersity shows some effects on asymmetry factor with REs reaching almost 6.5 % for relatively small aggregates containing 100 monomers or less. Considering the significant uncertainties on BC parameters, a relative error of 5 % is well acceptable for most applications, so Fig. 7 furthermore validates that the average volume correction ratio and empirical relationships can be applied confidently to account for the effects of minor structures on the optical properties of

BC aggregates. Results for compact aggregates are similar, and will not be shown here.

Table 4 shows the bulk optical properties of BC aggregates with minor structures from direct DDA simulations and those from the empirical approximations, and the optical properties of ideal fractal aggregates are also given as a reference. A



lognormal size distribution with a geometric mean diameter and a standard deviation of 120 nm and 1.5 for aggregate sizes are used (Reddington et al., 2013). It is clear from Table 4 that the bulk optical properties of ideal aggregates are significantly less than those of aggregates with minor structures, especially for MEC and MAC. With an average over the aggregate size, the empirical approximation should give even better approximations. The bulk MEC and MAC are

approximately 7 and 6 $m^2$ $g^{-1}$, respectively, at 500 nm, which is in reasonable agreement with observations (Bond and Bergstrom, 2006; Kahnert et al., 2010; Liu et al., 2010; Xu et al., 2017). The empirical results overestimate MEC and MAC of polydispersity by 2.8 % and 3.7 %, respectively, and underestimate those of necking by 4.5 % and 3.9 %. Again, minor structures show less impact on the bulk asymmetry factor and SSA, and the relative errors from those of idealized aggregates are always less than 2 %. Clearly, after the scaling with the empirical relationship, results from the perfect aggregates can be

readily used to represent the bulk optical properties of aggregates with minor structures.

The results discussed above are at a wavelength of 500 nm, and results at different incident wavelengths between 300 and 700 nm are illustrated in Fig. 8. Aggregates with 200 monomers and the fractal dimension of 1.8 are considered for these simulations at a volume variation of 0.1. Obviously, all the optical properties decrease with the increase of wavelength,

because the particles become relatively smaller compared to the longer wavelength. The empirical relationship with the same correction ratio 1.05 is applicable for different wavelengths. Again, all the optical properties of BC aggregates with different minor structures (markers) are within the range of ± 5 % from the results of empirical approximations (the solid black lines).

Most previous studies as well as aforementioned results consider minor structures individually, i.e., the BC aggregates

deviate from the ideal ones by only one of the five structures, whereas atmospheric BC aggregates in general contain more than one structure simultaneously. To develop more realistic models to represent actual BC particles, we further investigate whether the effects of a combination of different minor structures can still be accounted by the simple empirical relationships. To this end, we include multiple minor structures in single aggregate, and use the volume variation to constrain the combination. For a given total volume variation, the contribution of each minor structure can be randomly generated.

Figure 9 shows ten examples of aggregates containing 15 monomers with random combinations of minor structures to represent their detail morphologies, and five examples of aggregates containing 100 monomers, which are more close to actual BC aggregates, are given in the bottom panel. Different combinations show clearly different geometries, though they all have the same total volume variation of 0.1 from the corresponding perfect aggregate.

Figure 10 compares the optical properties of aggregates with multiple minor structures calculated by the DDA (blue bars) and those from empirical approximations (yellow bars). For each case, twenty random aggregates are built and calculated, and the same volume variation of 0.1 is imposed for imperfect aggregates. The averaged optical properties as well as their variances (vertical red errors bars) are shown in the figure. Surprisingly, even with different combinations of minor structures, the variances are quite small indicating that the differences between the twenty random cases are minor. The





asymmetry factor shows the largest variance, which agrees with the results of ideal aggregates (Liu et al., 2015). Comparing the blue and yellow bars, we can find that the empirical results agree closely with the accurate calculations with all relative errors less than 3 %. Therefore, even for BC aggregates with multiple minor structures, their combined effects can still be accurately accounted for by the empirical relationships proposed in this study.

**4 Conclusion**

This study investigates the effects of different minor structures on the optical properties of BC aggregates, and develops a simple empirical treatment to account for their effects from the optical properties of the corresponding ideal fractal aggregates. The structures considered in this study include polydispersity, irregularity, coating, necking, and overlapping, all of which are for monomers, not the aggregate overall structure. The volume variation can be used to unify the effects of

10 different minor structures. Minor structures significantly affect the optical properties of BC aggregates with different fractal dimensions and aggregate sizes, and with the increase of wavelength, these effects become slightly weaker. Among the five minor structures, necking shows the strongest effect. The asymmetry factor as well as the scattering matrix is almost unaffected by these minor structures. For easy and accurate estimation of BC aggregates with these minor structures, a simple empirical relationship is developed to account for their effects on the optical properties based on a constant correction

ratio of 1.05 and volume variation from that of the corresponding ideal aggregate. Results show that the empirical relationships can accurately represent the optical properties of BC aggregates with minor structures. Additionally, the effects of multiple minor structures are found to be similar to those of a single structure with the same volume variation, and the empirical relation is also applicable.

For practical applications, the detailed parameters of fractal aggregate as well as the minor structures should be known, and such knowledge can be obtained by the analysis of aggregate images (Brasil et al., 2000; Chakrabarty et al., 2006). Among the minor structures, the monomer size polydispersity and overlapping have been most widely considered (Bescond et al., 2014; De Temmerman et al., 2014; Bourrous et al., 2018). However, with the proposed empirical relationship, only simple estimations of the volume or mass variation are sufficient to account for the change in optical properties. The methodology

proposed in this study enables efficient and accurate prediction of the optical properties of BC aggregates with minor structures based on those of the corresponding ideal aggregates that can be calculated using the MSTM and GMM methods with much less computational efforts than the DDA, and the optical properties of those ideal aggregates have been given from a comprehensive database (Liu et al., 2019). This will make the applications more practical, and are instructive for experimental analysis.



## Acknowledgements

We are deeply thankful to Drs. Maxim A. Yurkin and Alfons G. Hoekstra for the ADDA code. This work was financially supported by the National Key Research and Development Program of China (2016YFA0602003), the National Natural Science Foundation of China (41505018), and the Young Elite Scientists Sponsorship Program by CAST (2017QNRC001). 5 The computation of this study was supported by the National Supercomputer Center in Guangzhou (NSCC-GZ).

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





**Table 1.** Comparison of the previous studies on effects of minor structures on optical properties of fractal aggregates.

| Reference | Minor structure | Parameters | | | Amplification factor |
| --- | --- | --- | --- | --- | --- |
| | | $N$ | Wavelength (nm) | Structure scale | |
| Farias et al., 1996 | Polydispersity | 16 - 256 | 400 - 700 | 1 - 2 | 1.0 - 2.2 |
| Liu et al., 2012 | Coating | 200 | 628 | 0.01 - 0.2 | 1.03 - 1.05 |
| Scarnato et al., 2013 | Coating | 64 - 100 | 200 - 1000 | 1.34 - 1.68 | 1.0 - 1.5 |
| Skorupski & Mroczka, 2014 | Necking Overlapping | 2 - 5 | 400 - 800 | 0.0 - 1.0 0.0 - 0.98 | 0.9 |
| Yon et al., 2015 | Necking Overlapping | 63 - 233 | 266 - 1064 | 0.005 - 1.0 0.0 - 0.4 | ~2 |
| Doner & Liu, 2015 | Overlapping Polydispersity | 20 - 103 | 532 - 1064 | 20 % 20 % | ~3.6 |
| Dong et al., 2015 | Coating | 50 - 600 | 550 | 0.0 - 0.85 | 1.15 - 2 |
| Doner et al., 2017 | Necking Overlapping Coating | 200 | 440 - 1020 | 0.05 - 0.5 0.002 - 0.2 0 % - 100 % | 1.15 |





**Table 2**. Relationship between different minor structure parameters ($c_i$ with $i$ from 1 to 5) and unified parameter volume variation $\beta$.

| Minor structures | | Parameters | $\beta$ | | | | | | |
|---|---|---|---|---|---|---|---|---|---|
| | | | 0 | 0.025 | 0.05 | 0.075 | 0.1 | 0.15 | 0.2 |
| Polydispersity | $M_1$ | $c_1$ | 1.0 | 1.090 | 1.130 | 1.136 | 1.160 | 1.194 | 1.225 |
| Irregularity | $M_2$ | $c_2$ | 1.0 | 0.985 | 0.970 | 0.955 | 0.935 | 0.910 | 0.880 |
| Coating | $M_3$ | $c_3$ | 1.0 | 1.009 | 1.017 | 1.025 | 1.033 | 1.048 | 1.064 |
| Necking | $M_4$ | $c_4$ | 0.0 | 0.502 | 0.594 | 0.655 | 0.702 | 0.772 | 0.825 |
| Overlapping | $M_5$ | $c_5$ | 1.0 | 1.008 | 1.017 | 1.025 | 1.033 | 1.049 | 1.063 |





**Table 3.** Correction ratio obtained from results in Fig. 4 to give the absorption and scattering cross sections of aggregates with different minor structures.

| Minor structure | $B_{ABS}$ | $B_{SCA}$ |
|---|---|---|
| Polydispersity | 1.02 | 1.01 |
| Irregularity | 1.05 | 1.04 |
| Coating | 1.05 | 1.04 |
| Necking | 1.08 | 1.06 |
| Overlapping | 1.03 | 1.03 |



**Table 4.** Bulk scattering properties for lacy aggregates with minor structures ($\beta$ =0.1) of direct simulations and those from perfect aggregate structures but with corrections. The values in the parentheses are the relative errors (%) from the empirical results.

|  | MEC ($m^2\ g^{-1}$) | MAC ($m^2\ g^{-1}$) | SSA | $g$ |
|---|---|---|---|---|
| Ideal | 6.87 | 5.90 | 0.12 | 0.54 |
| Polydispersity | 6.95 (2.8) | 5.91 (3.7) | 0.15 (< 0.1) | 0.55 (-1.8) |
| Irregularity | 7.03 (1.7) | 5.98 (2.5) | 0.15 (< 0.1) | 0.55 (-1.8) |
| Coating | 7.12 (0.4) | 6.03 (1.6) | 0.15 (< 0.1) | 0.54 (< 0.1) |
| Necking | 7.49 (-4.5) | 6.38 (-3.9) | 0.15 (< 0.1) | 0.54 (< 0.1) |
| Overlapping | 7.11 (0.6) | 6.03 (1.7) | 0.15 (< 0.1) | 0.54 (< 0.1) |
| Empirical | 7.15 | 6.13 | 0.15 | 0.54 |




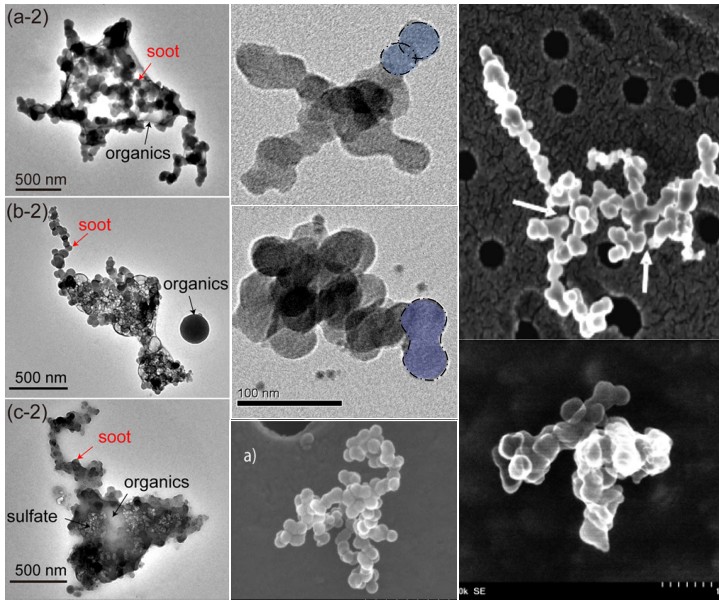

**Figure 1.** Examples of TEM/SEM images of BC particles from different in situ or laboratory observations (Gwaze et al., 2006; Kamimoto et al., 2007; Chakrabarty et al., 2009; Yon et al., 2015; Wang et al., 2017).



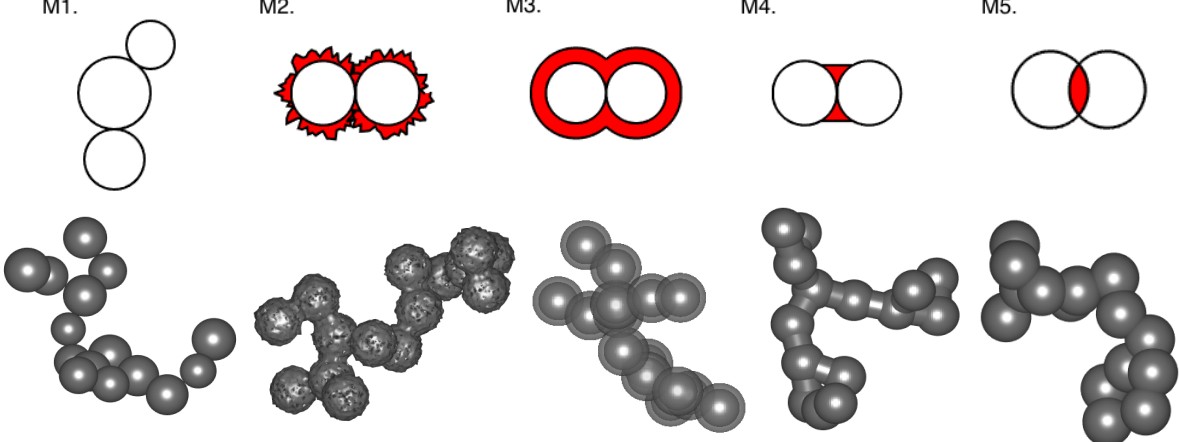

**Figure 2.** Two- and three-dimensional models of five minor structure. Aggregates with 15 monomers and a fractal dimension of 1.8 are considered as an example.





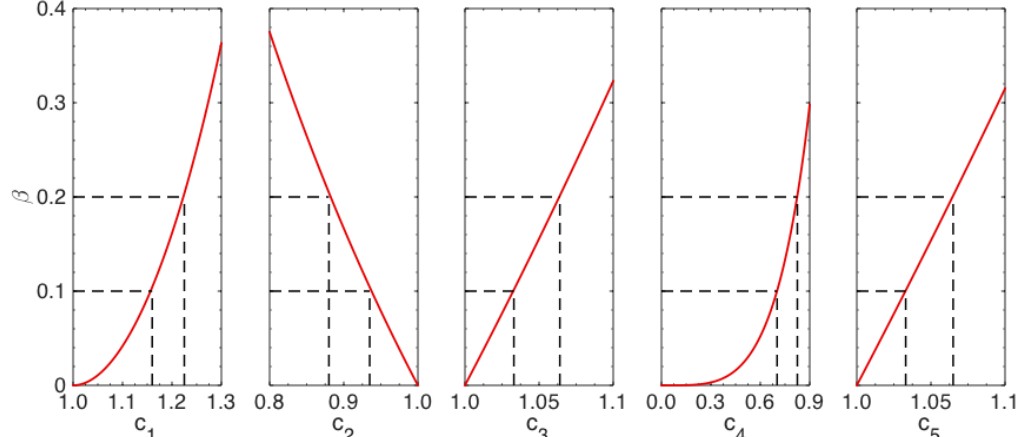

**Figure 3.** Relationships between geometry parameters and volume variations.



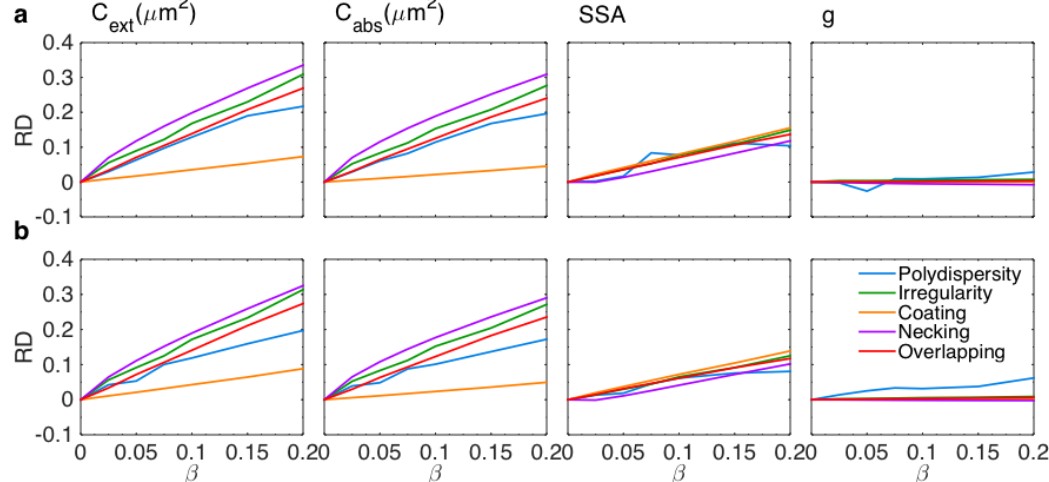

**Figure 4.** Relative differences of optical properties for aggregates with minor structures compared to those with perfect fractal aggregate structures at a wavelength of 500 nm. Aggregates with 200 monomers and a fractal dimension of 1.8 (top panels) and 2.3 (bottom panels) are considered.




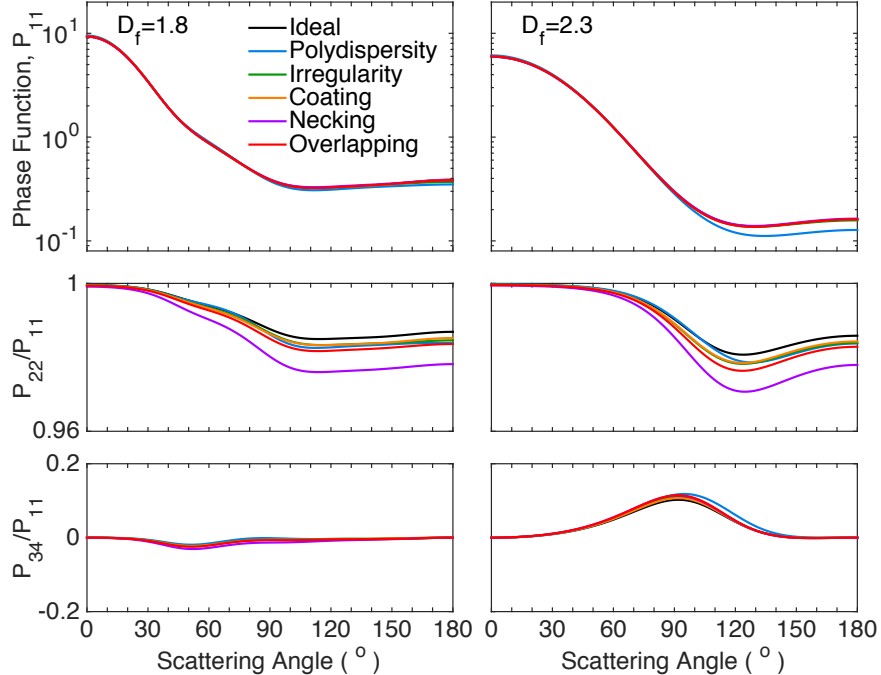

**Figure 5.** Three normalized scattering matrix elements ($P_{11}$, $P_{12}$, and $P_{34}$) of fractal aggregates with minor structures.



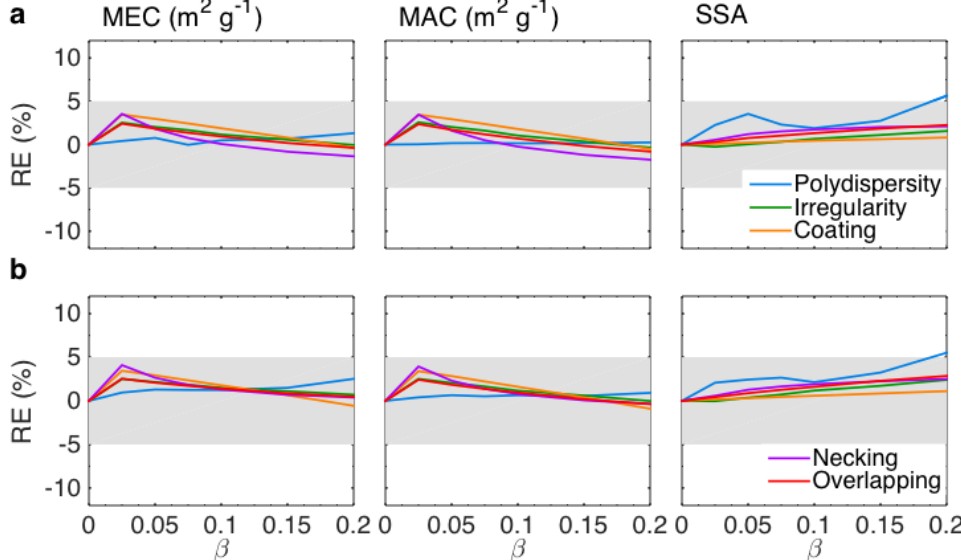

**Figure 6.** Relative errors of optical properties for aggregates with minor structures between direct DDA simulations and those modified from ideal fractal aggregates using the empirical correction factors and volume/mass variation.

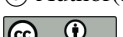



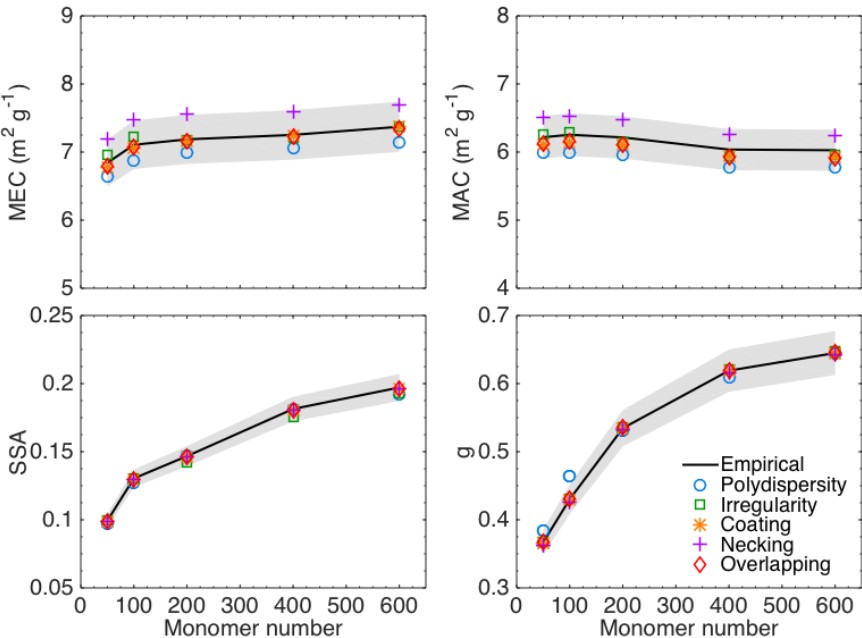

**Figure 7.** Comparison of the optical properties of aggregates with minor structures from direct simulations (markers) and those from ideal aggregates but with corrections to accounting for the minor structures (black lines).





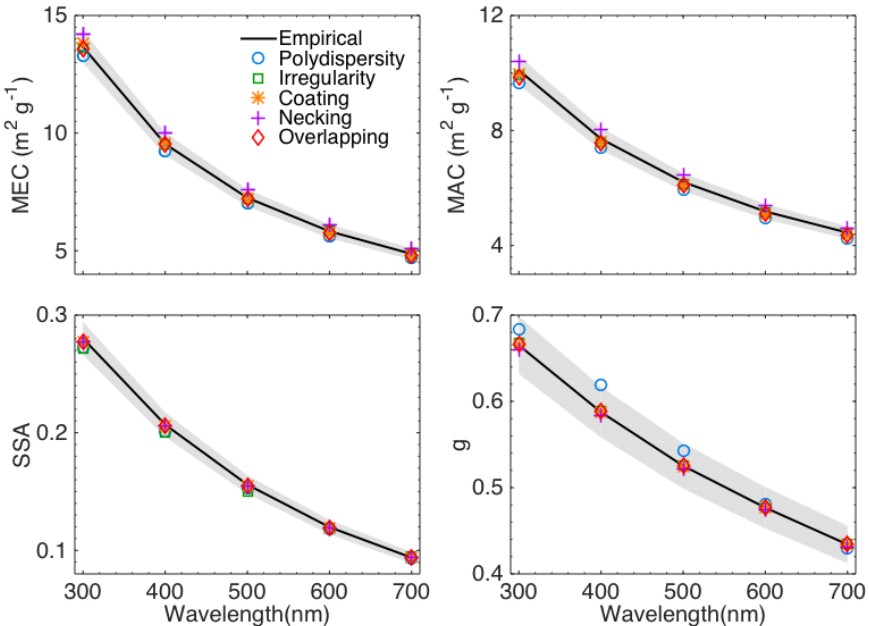

**Figure 8.** Same as Fig. 7 but for optical properties of aggregates with 200 monomers at different wavelengths.





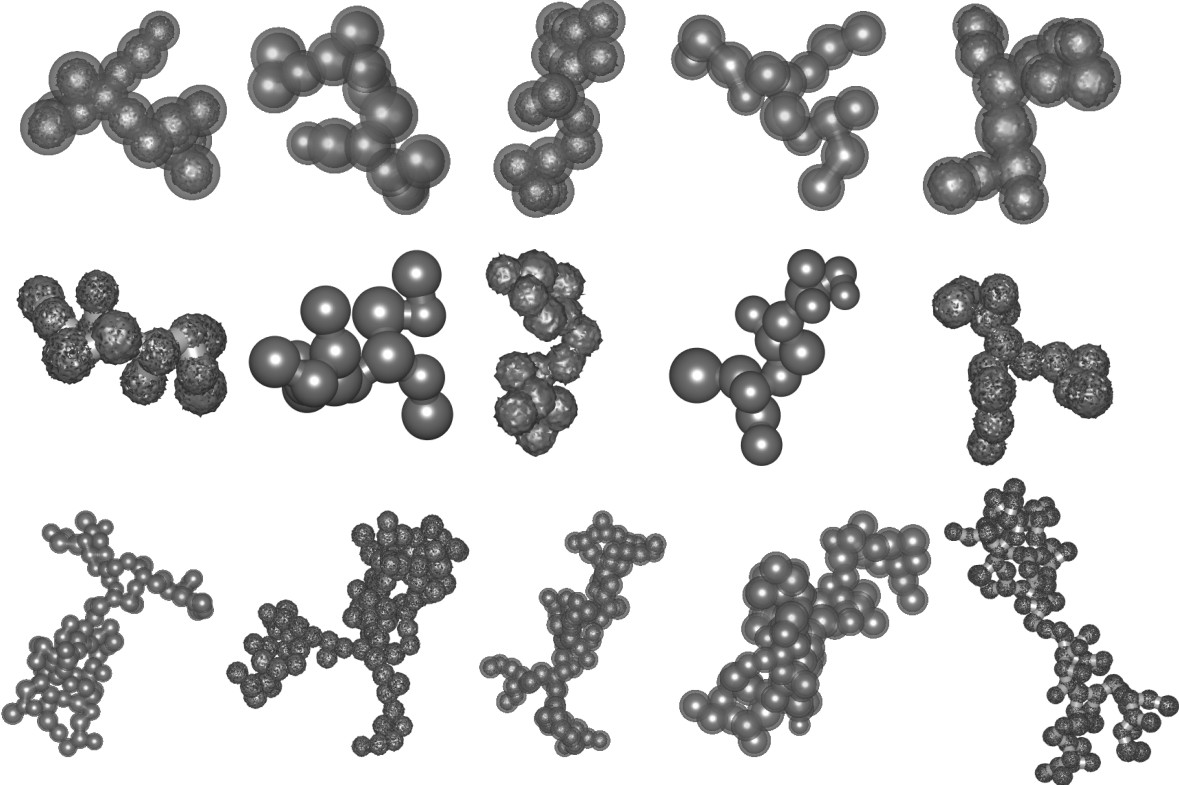

**Figure 9.** Examples of aggregates with random combinations of minor structures. The aggregates are generated to have the same volume variation of 0.1 compared to the perfect counterparts.





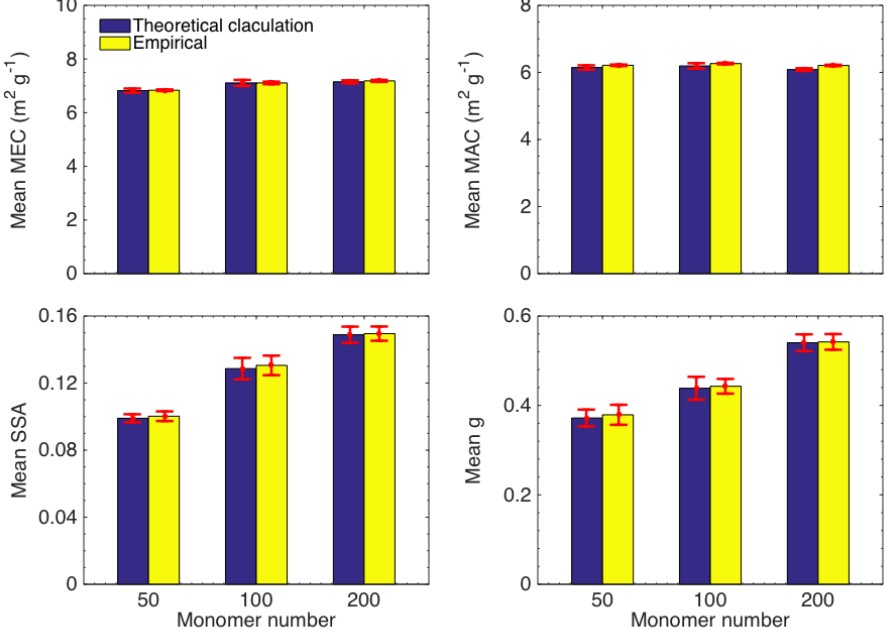

**Figure 10.** Comparison of the optical properties from direct simulations of aggregates with combined multiple minor structures (blue) and those from perfect aggregate structures but with corrections to accounting for the minor structures (yellow).