# Peer review of "Accounting for the effects of non-ideal minor structures on the optical properties of black carbon aerosols"

_Atmospheric Chemistry and Physics, 2018_

## Referee Comment (RC1) · Anonymous Referee #1 · 12 Dec 2018

This paper studies the optical properties of black carbon particles and introduces a new parameter, the "volume variation" to quantify several minor structural differences relative idealized structures such as fractal aggregates of same sized, spherical primary particles, etc.

This paper has some admirable properties. It provides a thorough review of the light scattering literature of aggregates such as soot and emphasizes the non-ideality of their real world structures. It also classifies the several ways structures can deviate from the ideal form. The results demonstrate that all these non-idealities can by represented by a volume variation that can be used to unify their effects. Then a simple empirical

relationship quantifies their effects on the optical properties. Overall the effects are not large, a few to several percent. The authors make arguments that such effects can be important. Important or not, it is worthwhile to know the extent of the effects and compare them as this paper does.

The paper is well written and the results are of value. I recommend publication.

---

## Referee Comment (RC2) · Anonymous Referee #2 · 19 Dec 2018

General comment: This paper addressed the effects of minor geometric structures of the black carbon particles to their optical properties using DDA simulations, in a more detailed-and-comprehensive manner than any previous publications. I can recommend publication considering the huge efforts and the excellent quality of writing. However, I have some questions and critical comments on the methods and the result interpretations which should be taken into account in the paper before its publication.

Major comments: (1) The considered size range of BC particles in the DDA simulations seems to be Dv (volume equivalent diameter) < ~250 nm (at a=15nm, N=600), which could be a minor fraction of the whole size range of ambient BC particles. If the authors

suggest the significance/usefulness of their results for radiative forcing estimations and aerosol remote sensings (page 1. line 15), they should provide appropriate rationale on the considered size-range. It is not evident whether the "correction ratio" defined and evaluated by the authors also applies to the BC particles not smaller than the wavelength.

(2) If the authors include any quantitative interpretations of the DDA-simulated optical properties with the accuracy of the order of several percents, they need to include careful evaluations of the absolute accuracy of their DDA results. The surface granularity inherent to the DDA model is known to cause systematic overestimation of the absorption cross-section for Rayleigh-sized particles [Draine 1988], which is persistent even in the long-wavelength limit. This is one of the major reasons for the difficulty of modeling the soot optical properties using the DDA [Yurkin 2007; Moteki 2016]. I can suggest a quick evaluation of the systematic error of DDA results presented in this paper by comparing the exact Mie solution and DDA calculation (using ∼900 dipoles) for an isolated monomer. To my feeling, it might be difficult to separate the effects of minor structure M2 and the DDA-artifact unless each monomer is represented by a huge number of dipoles (> ∼10000).

(3) In general, particle's orientation relative to the propagation direction of the incident wave substantially affects the optical properties (e.g., absorption cross-section) for a fractal-like cluster of spheres. Please clarify how the authors treat/assume the particle's orientation in their DDA simulations because the derived "correction ratio" might also change depending on the assumed orientation.

(4) The authors assumed "1.8+0.6i" for the refractive index of BC throughout this paper. However, this parameter is still highly uncertain [Bond and Bergstrom 2006] and might vary depending on the emission source. The authors need to provide appropriate reasons for choosing this value and explain the expected consequence of the assumption.

Minor/technical comments: (1) page 3 line 16: Please clarify that the "266 nm" means wavelength.

References Yurkin, Maxim A., and Alfons G. Hoekstra. "The discrete dipole approximation: an overview and recent developments." Journal of Quantitative Spectroscopy and Radiative Transfer 106.1-3 (2007): 558-589.

Draine, Bruce T. "The discrete-dipole approximation and its application to interstellar graphite grains." The Astrophysical Journal 333 (1988): 848-872.

Moteki, Nobuhiro. "Discrete dipole approximation for black carbon-containing aerosols in arbitrary mixing state: A hybrid discretization scheme." Journal of Quantitative Spectroscopy and Radiative Transfer 178 (2016): 306-314.

Bond, Tami C., and Robert W. Bergstrom. "Light absorption by carbonaceous particles: An investigative review." Aerosol science and technology 40.1 (2006): 27-67.
* * *

---

## Referee Comment (RC3) · Anonymous Referee #3 · 28 Dec 2018

This manuscript deals with non-ideal minor structures effect to optical properties of black carbon aerosols. The subject falls clearly to the scope of ACP and it presents new correction factor to account for the mass/volume normalized absorption and scattering of non-ideal aggregates in comparison to ideal ones. The manuscript is well written and it proceesd in logical manner, and it thoroughly enough explains the used methods and outcomes. The title reflects the content of the paper and the abstract provides complete summary. I recommend accept the manuscipt with minor revisions with followig consideration. What I am missing are the examples how this new factor would change e.g. radiative transfer calculations (radiative forcing) or analysis of the experimental measurements compared to present estimations. I highly recommend to

add such examples.

---

## Author Comment (AC1) · 28 Jan 2019

**Responses to Reviewers (ACP Manuscript # ACP-2018-1102)**

First of all, we would like to thank the editor and three anonymous reviewers for their thoughtful review and valuable comments to the manuscript. In the revision, we have accommodated all the suggested changes into consideration and revised the manuscript accordingly. All changes are highlighted in RED in the revision. In this point-to-point response, the reviewers' comments are copied as texts in BLACK, and our responses are followed in BLUE.

**Anonymous Referee #1**

This paper studies the optical properties of black carbon particles and introduces a new parameter, the "volume variation" to quantify several minor structural differences relative idealized structures such as fractal aggregates of same sized, spherical primary particles, etc.

This paper has some admirable properties. It provides a thorough review of the light scattering literature of aggregates such as soot and emphasizes the non-ideality of their real world structures. It also classifies the several ways structures can deviate from the ideal form. The results demonstrate that all these non-idealities can by represented by a volume variation that can be used to unify their effects. Then a simple empirical relationship quantifies their effects on the optical properties. Overall the effects are not large, a few to several percent. The authors make arguments that such effects can be important. Important or not, it is worthwhile to know the extent of the effects and compare them as this paper does.

The paper is well written and the results are of value. I recommend publication.
**Response**: We really appreciate the reviewer's recognition of the scientific merit of this study. We agree with the reviewer that it is important to understand quantitatively the effects of the non-ideal minor structures and to compare them.
More importantly, we agree with the reviewer to interpret and to explain 'the effects' more carefully. The manuscript shows that the effects caused by minor structure are mainly contributed by the volume differences, and, after removing the influence of volume, the effects on the scattering and absorption are in the order of a few percent. Thus, we conclude that, in future studies, the understanding and evaluation of the particle's volume are more important than those on the minor structures themselves, and our empirical treatment can be used to account for such effects efficiently. To better present these conclusions, we improved the discussions in the abstract (Lines 26-33 on Page 1) and conclusion (Lines 25-28 on Page 13) section, and also added a paragraph in Section 3 to present a more quantitative example to clarify the conclusions on radiative forcing simulations (starting from Line 25 on Page 12).

---

## Author Comment (AC2) · 28 Jan 2019

**Responses to Reviewers (ACP Manuscript # ACP-2018-1102)**

First of all, we would like to thank the editor and three anonymous reviewers for their thoughtful review and valuable comments to the manuscript. In the revision, we have accommodated all the suggested changes into consideration and revised the manuscript accordingly. All changes are highlighted in RED in the revision. In this point-to-point response, the reviewers' comments are copied as texts in BLACK, and our responses are followed in BLUE.

**Anonymous Referee #2**

**General comment:**

This paper addressed the effects of minor geometric structures of the black carbon particles to their optical properties using DDA simulations, in a more detailed-and-comprehensive manner than any previous publications. I can recommend publication considering the huge efforts and the excellent quality of writing. However, I have some questions and critical comments on the methods and the result interpretations which should be taken into account in the paper before its publication.

**Response**: Thanks the reviewer for the constructive comments. The comments on the methods as well as the result interpretations significantly improve the quality of the manuscript, and make the paper more solid. The following presents our point-to-point responses as well as the revision for the manuscript.

**Major comments:**

(1) The considered size range of BC particles in the DDA simulations seems to be Dv (volume equivalent diameter) < ~250 nm (at a=15nm, N=600), which could be a minor fraction of the whole size range of ambient BC particles. If the authors suggest the significance/usefulness of their results for radiative forcing estimations and aerosol remote sensings (page 1. line 15), they should provide appropriate rationale on the considered size-range. It is not evident whether the "correction ratio" defined and evaluated by the authors also applies to the BC particles not smaller than the wavelength.

**Response**: Thanks for the suggestion. The reviewer mentioned an important factor for optical property simulations and result discussions. BC aerosols in the ambient atmosphere does show a quite wide range of size distributions (e. g. Schnaiter et al., 2005; Reddington et al., 2013). For the simulations of bulk optical properties in this study (Table 4), we considered aggregate equivalent volume diameter ($D_v$) to following a lognormal size distribution with a geometric mean diameter of 120 nm (Alexander et al., 2008; Chung et al., 2012). With this size distribution, most BC particles have $D_v$ between 50 and 300 nm, so we extended the size range up to ~ 350 nm (N up to 1500) in the revision. Thus, we updated the simulations for Table 4, and extended the results in Figure 8 to include much larger particles.

Furthermore, our results indicate that the effects of minor structures are less sensitive to particle sizes, so we think the size range considered is large enough.

Last but not least, although the DDA method is flexible on arbitrary particle shape, its computational efficiency is known to be relatively low as the particle size becomes large. Meanwhile, the MSTM method (Multiple-sphere T-matrix method) can't handle particles with minor structures but is reasonably efficient at larger size range for aggregates with perfect spheres. Thus, with the conclusion from this work, we can obtain the optical properties of

aggregates with minor structures more efficiently which is of great importance. This is discussed in the conclusion section (Line 1 of Page 14).

(2) If the authors include any quantitative interpretations of the DDA-simulated optical properties with the accuracy of the order of several percents, they need to include careful evaluations of the absolute accuracy of their DDA results. The surface granularity inherent to the DDA model is known to cause systematic overestimation of the absorption cross-section for Rayleigh-sized particles [Draine 1988], which is persistent even in the long-wavelength limit. This is one of the major reasons for the difficulty of modeling the soot optical properties using the DDA [Yurkin 2007; Moteki 2016]. I can suggest a quick evaluation of the systematic error of DDA results presented in this paper by comparing the exact Mie solution and DDA calculation (using ~900 dipoles) for an isolated monomer. To my feeling, it might be difficult to separate the effects of minor structure M2 and the DDA-artifact unless each monomer is represented by a huge number of dipoles (> ~10000).

**Response**: We agree with the reviewer that it is necessary to evaluate the accuracy of the method used for quantitative interpretation. The surface granularity inherent to the DDA model may limit the accuracy of the DDA simulations due to the small size scale of monomers and minor structures. Actually, we developed a systematic and comprehensive study to evaluate the performance of DDA on simulating the optical properties of BC aggregates by comparing with the MSTM results, and the results are published in a more technical paper in JQSRT (Liu et al., 2018). Therefore, we didn't re-evaluate the accuracy of the DDA method in this manuscript. Liu et al. (2018) showed that the DDA method shows relative errors less than 5% in the case of parameters used in this manuscript (i.e. the dpl of 200 and the refractive index of 1.8+0.6i), and such accuracy is enough for our discussions. Furthermore, the relative errors introduced by the DDA are systematic, while our work focus on the relative differences of optical properties between aggregates with minor structures and ideal aggregates. Thus, the systematic errors of the DDA will not influence the relative differences as well as our conclusions. We added some discussions in the revision to explain the accuracy of the DDA (Line 28 of Page 7).

Furthermore, the following figure illustrates the two-dimensional structures of actual isolated monomer (a and d) and those discretized by the DDA (b, c, e, f). (b) and (e) is for dpl of 200, and (c) and (f) is for dpl of 800. Comparing (b) and (e), the difference of surface roughness from perfect sphere mainly reflects on the protruding dipoles at the edge. It is clear from the figure that the structure of particle can be more accurately represented with the increase of dpl. The volume differences between (e) and (f) due to different dpls are less than 1%, and simulations show that the differences between its optical properties are also less than 1%, which can be ignored. Therefore, although the shape is not exactly represented with a dpl of 200, the overall structure can be captured and the results are almost not influenced. Considering the computational efficiency as well as the accuracy, we used a dpl of 200 for all simulations in this study, and thick this is enough for our results.

Actually, even smaller dpl values may be used for our study, because all minor structures are unified and defined by the "volume variance". We used the dipole number to indicate the exact particle volume in the DDA simulations, so the volume variance can be accurately tracked. In this way, even if the particle shape is not exactly presented by the dipoles, the results will not be significantly influenced as long as the overall structures are captured.

[Figure]

Figure R1. The discretization of the irregular spheres using different dipole sizes. (b) and (e) use a dpl of 200, and (c) and (f) use a dpl of 800.

(3) In general, particle's orientation relative to the propagation direction of the incident wave substantially affects the optical properties (e.g., absorption cross-section) for a fractal-like cluster of spheres. Please clarify how the authors treat/assume the particle's orientation in their DDA simulations because the derived "correction ratio" might also change depending on the assumed orientation.

**Response**: Thanks for the comment, and we did forget to clarify the particle orientation in the original manuscript. Particle orientation does slightly affect its optical properties, and, for aggregates, our previous study indicates that the effects caused by particle orientation are mostly less than 10% and compact aggregates are less sensitive to orientation (Liu et al., 2018). For practical applications, almost all numerical studies on optical properties, especially those of aerosols, consider atmospheric particles to be randomly oriented in the ambient atmosphere, and this becomes almost a default setting. Thus, all results presented in this study are those for randomly oriented particles, and we presented results averaged over different particle orientations. In the DDA simulations, the optical properties of a certain number of random particle orientation are simulated and averaged, and the results converge when the number of orientation up to a few tens (Liu et al., 2018). In the revision, we clarified and emphasized that all results are those for randomly oriented particles (see Lines 7 to 9 of Page 8).

(4) The authors assumed "1.8+0.6i" for the refractive index of BC throughout this paper. However, this parameter is still highly uncertain [Bond and Bergstrom 2006] and might vary depending on the emission source. The authors need to provide appropriate reasons for choosing this value and explain the expected consequence of the assumption.

**Response**: Thanks for the suggestion. As one of the most important parameters influencing the BC optical properties, the refractive index is also one of the most uncertain physical properties, because it can't be directly observed. It is difficult (or impossible) to find a single 'accurate' value to represent BC refractive index, so we just used a typical value in the manuscript. However, the reviewer mentioned an important factor (i.e., refractive index) that should be considered in the study. We added a sensitivity study to discuss the influence of refractive index on our results and conclusions, and the results are presented in the new Figure 7 as well as the corresponding discussions. The upper panels show BC optical properties as a function of the real part of refractive indices, and the lower ones are those as a function of the imaginary part. The refractive indices show clear effects on the optical properties, whereas the relative differences caused by the minor structures don't change too much. This means that the refractive index will change neither our conclusions nor the empirical relationship derived, and this ensures the application of our work for a wider range of BC refractive indices. The changes in the revision can be found in Lines 24 of Page 10 and Figure 7.

**Minor comments:**

(1) page 3 line 16: Please clarify that the "266 nm" means wavelength.
**Response**: Thanks, we have clarified (Line 16 of Page 3).

Reference:
[1] Alexander, D. T. L., Crozier, P. A. and Anderson, J. R.: Brown carbon spheres in East Asian outflow and their optical properties, Science, 321, 833-836, 2008.
[2] Chung, C. E., Lee, K. and Müller, D.: Effect of internal mixture on black carbon radiative forcing, Tellus B: Chemical and Physical Meteorology, 64, 10925, doi: 10.3402/tellusb.v64i0.10925, 2012.
[3] Liu, C., Teng, S., Zhu, Y., Yurkin, M. A., and Yung, Y. L.: Performance of the discrete dipole approximation for optical properties of black carbon aggregates, J. Quant. Spectrosc. Radiat. Transfer, 221, 98-109, 2018.
[4] Reddington, C. L., McMeeking, G., Mann, G. W., Coe, H., Frontoso, M. G., Liu, D., Flynn, M., Spracklen, D. V., and Carslar, K. S.: The mass and number size distribution of black carbon aerosol over Europe, Atmos. Chem. Phys., 13, 4917-4939, 2013.
[5] Schnaiter, M., Linke, C., Möhler, O., Naumann, K.-H., Saathoff, H., Wagner, R., and Schurath, U.: Absorption amplification of black carbon internally mixed with secondary organic aerosol, J. Geophys. Res., 110, D19204, doi:10.1029/2005JD006046, 2005.

---

## Author Comment (AC3) · 28 Jan 2019

**Responses to Reviewers (ACP Manuscript # ACP-2018-1102)**

First of all, we would like to thank the editor and three anonymous reviewers for their thoughtful review and valuable comments to the manuscript. In the revision, we have accommodated all the suggested changes into consideration and revised the manuscript accordingly. All changes are highlighted in RED in the revision. In this point-to-point response, the reviewers' comments are copied as texts in BLACK, and our responses are followed in BLUE.

**Anonymous Referee #3**

This manuscript deals with non-ideal minor structures effect to optical properties of black carbon aerosols. The subject falls clearly to the scope of ACP and it presents new correction factor to account for the mass/volume normalized absorption and scattering of non-ideal aggregates in comparison to ideal ones. The manuscript is well written and it proceeds in logical manner, and it thoroughly enough explains the used methods and outcomes. The title reflects the content of the paper and the abstract provides complete summary. I recommend accept the manuscript with minor revisions with following consideration. What I am missing are the examples how this new factor would change e.g. radiative transfer calculations (radiative forcing) or analysis of the experimental measurements compared to present estimations. I highly recommend to add such examples.

**Response**: Thanks the reviewer for the positive comments on the manuscript, and the constructive suggestion makes this work more complete.

In the revision, we added an example to discuss the effects of minor structures on the radiative forcing simulations (starting from Line 25 of Page 12). To better explain the conclusion of this work and the effects of minor structures, three cases are designed to calculate BC radiative forcing: (1). BC aggregates with minor structures and a volume variance from the ideal case of 10%; (2). BC aggregates with ideal aggregate structures; and (3). BC aggregates with ideal aggregate structures but the same total mass as those for the non-ideal case (i.e., Case (1)). As expected, the effects of minor structures on radiative forcing are similar to those on the optical properties, and the influences are also mainly caused by the changes on aggregate total volume/mass. Meanwhile, we emphasized in the conclusion section that the importance of this study is not only to evaluate and to unify the effects of minor structures, but also to present an efficient empirical relationship to account for their effects. Whatever the effects are interpreted, the effects of minor structures are easily accounted for without the tedious simulations of the optical properties for particles with minor structures or without even knowing their details.

An example of how these BC minor structures would influence the interpretation of experimental studies can hardly be given, as to date the measurement accuracy in terms of the MAC and MEC are not good enough and at best in the 5% range that is expected for the minor structures. However, the measurement capabilities are continuously improving mainly in terms of optical detection sensitivity, particle mass determination, as well as measurement comprehensiveness (e.g. by including size-segregated and spectrally resolved measurements as well as by adding detailed microscopic analysis of the particle morphologies). Therefore, we are confident that our study will be used in future to interpret the results of such detailed laboratory studies and the remaining differences when comparing with fractal particle light scattering models.